# Vpr and Its Cellular Interaction Partners: R We There Yet?

**DOI:** 10.3390/cells8111310

**Published:** 2019-10-24

**Authors:** Helena Fabryova, Klaus Strebel

**Affiliations:** Laboratory of Molecular Microbiology, National Institute of Allergy and Infectious Diseases, National Institutes of Health, Bldg. 4, Room 312, 4 Center Drive, MSC 0460, Bethesda, MD 20892, USA; helena.fabryova@nih.gov

**Keywords:** HIV-1, Vpr, accessory genes, host restriction factors

## Abstract

Vpr is a lentiviral accessory protein that is expressed late during the infection cycle and is packaged in significant quantities into virus particles through a specific interaction with the P6 domain of the viral Gag precursor. Characterization of the physiologically relevant function(s) of Vpr has been hampered by the fact that in many cell lines, deletion of Vpr does not significantly affect viral fitness. However, Vpr is critical for virus replication in primary macrophages and for viral pathogenesis in vivo. It is generally accepted that Vpr does not have a specific enzymatic activity but functions as a molecular adapter to modulate viral or cellular processes for the benefit of the virus. Indeed, many Vpr interacting factors have been described by now, and the goal of this review is to summarize our current knowledge of cellular proteins targeted by Vpr.

## 1. Introduction

Primate lentiviruses are a group of retroviruses that cause slowly progressing, often incurable diseases. A characteristic feature of these viruses is the presence of accessory genes that vary in number from virus to virus (Figure 1). The least complex lentivirus is Equine Infectious Anemia Virus (EIAV), which aside from the prototypic *gag*, *pol*, and *env* genes, encodes only a single accessory gene, *S2*, that is not found in any of the other lentiviruses. Interestingly, however, the EIAV *S2* gene product appears to be a functional homolog of the Nef protein encoded by human and simian lentiviruses in that it has the ability to counteract the antiviral activity of the host factors SERINC3 and SERINC5 [1]. At the other end of the complexity spectrum are the human and simian immunodeficiency viruses, which encode four accessory genes each, including *vif*, *vpr*, *vpx*, *vpu*, and *nef*. Of those, *vif*, *vpr*, and *nef* are encoded by almost all HIVs and SIVs, while *vpu* and *vpx* genes are found only in a limited set of these viruses (Figure 1). Some accessory open reading frames have also been described for the oldest integrations of lentiviruses. However, none of these open reading frames has been assigned to Vpr indicating that *vpr* is a more recently acquired accessory gene [2,3].

The function of the various lentiviral accessory proteins has been extensively studied and it is now clear that none of them have enzymatic activity. Instead, all accessory proteins appear to function as molecular adaptors to connect or reprogram viral and cellular functions at least in part for the purpose of neutralizing innate host defense mechanisms (for review see [4]). While the research community has gained significant mechanistic insights into the functions of Vif, Vpu, Vpx, and Nef proteins, the role of Vpr in viral replication and pathogenesis has remained largely an enigma. Early on, Vpr was shown to enhance HIV-1 replication in T cells and was the first accessory protein to be found encapsidated into virions [5,6]. Indeed, while other HIV accessory proteins such as Vif and Nef seem to be packaged relatively nonspecifically without apparent interaction with structural viral proteins, studies on Vpr identified sequences in the P6 domain of the viral Gag precursor, as well as in Vpr, required for the specific packaging of Vpr into virions [7,8,9,10], which is facilitated by oligomerization [11,12,13]. Of note, Vpr is not packaged in equimolar amounts to Gag.

Indeed, quantitative analyses indicate that the ratio of Gag to HIV-1 Vpr is approximately 7:1 [14]. Assuming 5000 Gag molecules per virion, virus particles would contain about 700 Vpr molecules [5,14,15]. Other estimates for the amounts of HIV-1 Vpr packaged into virions are somewhat lower with approximately 275 molecules per virion [16]. HIV-2 was found to package even less Vpr (40–50 copies). However, these viruses additionally package large amounts of Vpx (2000–3000 copies) [17]. The specific packaging of Vpr through an interaction with the viral Gag precursor suggests a function for Vpr early during virus replication. However, trans-complementation studies did not reveal an effect of virus-associated Vpr in single-round infections of monocytes [18]. On the other hand, studies investigating the error rate of HIV-1 reverse transcriptase revealed a four-fold higher error rate in reverse transcripts in the absence of Vpr [19,20]. The reduced error rate in the presence of Vpr was dependent on it being packaged into viral particles, thus implying a role for virus-associated Vpr during the early phase of virus replication [19]. Furthermore, it was reported that Vpr inhibits the aminoacylation of tRNA^Lys^, thereby promoting tRNA^Lys^ packaging into viral particles where it is required for the initiation of reverse transcription [21]. It is interesting that in mature virions, Vpr remains associated with the viral core despite the quantitative loss of p6 from core preparations [14]. Cellular localization studies indicate that in infected cells, Vpr primarily localizes to the nucleus suggesting a role of Vpr in later stages of virus replication [22,23,24]. Despite all of these insights, some of which have been reported as much as 30 years ago, the precise role of virion-associated Vpr remains unclear. Part of the challenge in characterizing the function of Vpr has been the fact that its contribution to viral replication in most commonly used experimental tissue culture systems such as immortalized CD4+ T cell lines (e.g., A3.01, Jurkat, H9, or CEM-SS) is generally modest (see Figure 2A–D) [22,23]. On the other hand, studies first performed on HIV-2 identified a severe restriction of *vpr*-defective virus for virus replication in macrophages [25], an observation that was confirmed later for HIV-1 as well [18,26,27]. Indeed, our own studies confirmed a severe restriction of *vpr*-defective HIV-1 AD8 in differentiated THP-1 cells and terminally differentiated primary human monocyte-derived macrophages (Figure 2E,F). Interestingly, while HIV-1 Vpr was found to be crucial for the infection of macrophages, it did not facilitate the infection of non-dividing CD4+ T cells [28].

Studies performed in non-human primates provide a glimpse into the in vivo importance of Vpr. One study reported that SIVmac lacking a translation initiation codon in *vpr* replicated with similar kinetics as *vpr*-positive virus in vitro in macaque lymphocyte cultures as well as the human CEMx174 cell line. However, upon infection of rhesus macaques, the same *vpr*-defective virus rapidly reverted back to a functional ATG in more than half of the infected animals [29]. Similarly, virus isolated from a lab worker who had been accidentally infected with the *vpr*-defective HIV-1 IIIB isolate revealed reversion of the *vpr* ATG [30]. These observations suggest that Vpr may be critical for virus replication in vivo under certain circumstances but is dispensable for replication in most tissue culture systems. Another study, however, found no significant differences between wt and Vpr-null SIVmac replication in vivo and progression to AIDS [31].

Although the mechanism by which Vpr contributes to viral pathogenesis remains unclear, viruses with mutations in *vpr* are found to be abundant in long-term non-progressor patients [32], further suggesting that Vpr expression is important for virus replication in vivo.

Only recently have we begun to understand the possible role(s) of Vpr for virus replication at a mechanistic level. It is now appreciated that Vpr can assemble a CRL4^DCAF1^ E3 ubiquitin ligase complex mediated by the WD40 protein VprBP (DCAF1) [33,34] to target multiple cellular proteins for proteasomal degradation [35,36,37,38,39,40,41,42,43,44,45,46,47,48,49,50,51,52]. Many but not all of the proteins targeted by Vpr are involved in cellular DNA repair and cell cycle arrest [53]. Nevertheless, we still have a long way to go to fully understand the functional properties of this highly conserved lentiviral protein. It is generally assumed that, like all other lentiviral accessory proteins, Vpr functions as an adaptor molecule to interact not only with the P6 region of the Gag precursor but also with host factors to either induce their degradation or to otherwise modulate their activities. Thus, the recent focus in the Vpr field has been the identification and characterization of host factors targeted by Vpr. The current review aims at summarizing our current knowledge on Vpr and its functional interaction with host factors. Since much of the work on Vpr has been done on HIV-1 Vpr, Vpr here stands for HIV-1 Vpr unless otherwise specified.

## 2. Mutational Characterization of Vpr

Vpr (Viral Protein R) is a protein with an apparent molecular mass of about 15 kDa [54]. Vpr has been reported to be post-translationally modified by phosphorylation at three serine residues (S_79_, S_94_, S_96_), a feature that may be important for its biological activity [55]. Two of these serine residues (S_79_, S_96_) are highly conserved. Overall, however, Vpr reveals a significant degree of sequence polymorphism (www.hiv.lanl.gov). As mentioned above, polymorphisms in Vpr may be functionally significant as one study identified a correlation between Vpr polymorphisms and long-term non-progressor phenotypes (LTNP) [32]. One has to keep in mind, however, that many of the sequences deposited in the Los Alamos database are derived from integrated, potentially defective, proviral genomes rather than from replicating viruses. Therefore, correlations of sequence polymorphisms with functional properties need to be taken with a grain of salt unless supported by experimental data. Indeed, introducing mutations into the *vpr* gene and analyzing its functional consequences in vitro has been a common approach to investigating the function of the Vpr protein. It is therefore, not surprising that over the years, numerous Vpr mutants were constructed and their functional properties studied. A summary of the reported mutations and their functional properties is listed in Table 1. It is more than likely that other mutants not shown in this table were constructed and tested but escaped our attention. For this, we apologize.

## 3. Role of Vpr in Nuclear Import of Viral Preintegration Complexes

In terms of virus replication, one of the earliest reported functions of Vpr was its role in nuclear import of lentiviral preintegration complexes presumably by facilitating its docking to nuclear pores [60,76,77,78,79,80,81,82,83,84,85]. Such a function occurs before Vpr can be synthesized in the infected cell and would thus have to be performed by Vpr that is associated with the incoming virus particle. As mentioned above, HIV-1 virions contain several hundred copies of Vpr [14], which are present in the nucleoprotein complex (also referred to as the preintegration complex) [76,86]. Upon fusion with the plasma membrane and partial uncoating, the preintegration complex moves along cellular microfilaments to the nuclear pore [87] and eventually is transported into the nucleus. Recent advances in imaging technologies allow for the tracking of single particles as they enter a cell. From such experiments, it was concluded that a substantial amount of Vpr that is present in viral particles is released from the preintegration complex shortly after fusion of the virus with the target cell and accumulates in the nucleus in a diffuse pattern [88]. There is evidence that nuclear import of the HIV-1 preintegration complex is a slow process, and that part of the uncoating occurs directly at the nuclear pore [89]. However, mutational analyses of Vpr suggest that Vpr targeting to the nuclear pore, while improving HIV-1 replication in macrophages, is not absolutely required for virus replication in these cells [56]. Thus, the precise role of Vpr in the nuclear uptake of viral preintegration complexes remains unclear. Indeed, there are a number of additional factors for which a role in nuclear import of preintegration complexes was reported. For instance, imaging studies demonstrated that aside from Vpr, HIV-1 CA and the nuclear pore component Nup358 are important for nuclear targeting of the preintegration complex [89,90]. Furthermore, transportin-3 (TNPO3) [91] and the nuclear pore complex component nucleoporin 153 (NUP153) [92,93,94] were shown to be important for infection by HIV-1. Yet other studies suggest a role for integrase [86,95] as well as a central polypurine tract-central termination sequence (cPPT-CTS) in nuclear entry [96]. Finally, two nuclear envelope proteins, SUN1 [97] and SUN2 [98,99,100], were shown to affect the nuclear import of HIV-1 preintegration complexes. It is fair to say that the nuclear uptake of HIV-1 preintegration complexes is a complex process that is still not fully understood in all its mechanistic details. As far as Vpr is concerned, there is evidence that it has functions above and beyond the support of nuclear entry. Indeed, while immunocytochemical analyses typically show an accumulation of Vpr in the nucleus, there are significant amounts of Vpr in the cytoplasm. In fact, mutation of the nuclear export signal in Vpr demonstrated that nuclear export of Vpr is required for replication in macrophages [101] although it is not required for p6-mediated packaging into virions [65].

## 4. Vpr and Transcriptional Regulation

Integrated HIV-1 proviruses carry long terminal repeats (LTRs) at both ends, which have binding sites for several host transcription factors [102]. Most prominent are two recognition sites for NFκB and three binding sites for Sp1 all of which are located in the LTR region upstream of the transcription start site [103]. NFκB is a cellular transcription factor with broad effects on transcriptional regulation of cellular genes (reviewed in [104]). These include TRAF1, TRAF2, a cellular inhibitor of apoptosis (c-IAP)1, c-IAP2, Mcl-1, IEX-1L, Bcl-xL, and A1/Bfl-1, all of which are able to inhibit activation of caspases at various steps in the caspase pathway ([105] and references therein). Typically, NFκB is expressed in an inactive cytoplasmic form that consists of the p50/p65 heterodimeric NFκB complex and the 37 kDa inhibitor IκB, which prevents nuclear translocation of the p50/p65 complex. Phosphorylation of IκB at two conserved serine residues through upstream signal-induced kinases leads to proteasomal degradation of IκB by the SCF^TrCP^ complex and results in the translocation of NFκB to the nucleus and transcriptional activation of genes carrying NFκB binding sites [106,107,108]. Indeed, inhibition of NFκB activation by HIV-1 Vpu, which sequesters TrCP, thereby interfering with the TrCP-dependent degradation of IκB, was shown to drive HIV-infected cells into apoptosis [105,109]. The transcription factor Sp1 is considered a basal transcription factor whose main role appears to be the regulation of housekeeping genes. Interestingly, Sp1 has the ability to recruit transcription machinery in the absence of a TATA box [110]. With regard to HIV-1, both NFκB and Sp1 can act as transcriptional activators [111,112]. HIV-1 transcriptional activation through Sp1 was reported to involve a leucine-zipper motif in Vpr, which stabilizes the binding of multiple Sp1 transcription factors to the three Sp1 sites on the HIV-1 LTR [113,114]. This mode of action implies that Vpr may regulate not only transcription from the HIV LTR but impact expression of other genes that are under the control of Sp1, which indeed has been reported [115]. Transcriptional activation via NFκB was shown to be mediated through the p300 transcriptional co-activator, which promotes cooperative interactions between the Rel A subunit of NFκB and cyclin B1.Cdc2 [116]. In addition, HIV-1 Vpr was shown to stimulate the NFkB and AP-1 signaling through an upstream signaling pathway involving transforming growth factor-β-activated kinase 1 (TAK1) [73]. Vpr induces phosphorylation of TAK1, which results in TAK1 activation and ultimately NFκB activation [73]. However, Vpr was also shown to be able to act as a transcriptional repressor. For instance, Vpr was found to inhibit the NFκB pathway by upregulating IκB expression [117]. Furthermore, SIVcol and SIVolc Vpr were found to suppress NF-κB activation in a DCAF1-independent manner, leading to decreased production of IFNs in later stages of infection [118]. Other genes whose expression is regulated by Vpr are IFNA1 [119,120], glucocorticoid receptor [121,122], as well as IL-6 and IL-10 [123]. Finally, Vpr was implicated in the regulation of HIV latency as serum Vpr was able to activate virus expression in resting PBMC from HIV-infected individuals [124,125]. In addition, a recent study reports that CTIP2, a cellular factor associated with chromatin-modifying enzymes, contributes to HIV-1 gene silencing in latently infected cells but is degraded by Vpr via a Cul4A-DDB1-DCAF1 ubiquitin ligase complex [126]. Thus, Vpr may contribute to the reversal of HIV-1 latency in latently infected cells. Whether Vpr acts as a transcriptional activator or repressor may ultimately depend on the cellular milieu and the relative levels of Vpr-interacting activating or repressing factors.

## 5. Vpr Effect on Env Stability in Macrophages

A recent study investigating the role of Vpr in terminally differentiated macrophages found that, consistent with a previous report [18], Vpr had no effect on the first round of infection but significantly impacted de novo virus production following the initial infection [120]. In fact, it was noted that Vpr-deficient virions contained significantly lower amounts of Env protein, which was attributed to increased lysosomal degradation of Env [120]. A subsequent study found that Vpr increases Env expression in dendritic cells and CEM.NKR cells [127]. However, these authors suggest that Env degradation in the absence of Vpr occurs via endoplasmic reticulum-associated protein degradation (ERAD). Thus, while there is a consensus about the Env-stabilizing effect of Vpr, the mechanism of Env destabilization/degradation in the absence of Vpr remains under debate. And, of course, the mechanism by which Vpr affects Env expression remains unclear as well. One hypothesis is that macrophages express a cell type-specific restriction factor that targets newly synthesized Env [120]. Indeed, a recent study that was published as a non-peer-reviewed preprint in bioRxiv implicates mannose receptor (MR) as the macrophage-specific host factor [128]. MR expression was previously shown to be downmodulated in HIV-infected macrophages to counteract a BST2-like activity that interfered with the detachment of newly synthesized particles from the cell surface [129]. While previous studies implicated Nef and Tat in the downmodulation of MR [130,131], the most recent study reports an involvement of Vpr. Indeed, Vpr appeared to be sufficient for the efficient inhibition of MR expression in productively infected macrophages [128]. Surprisingly, silencing of MR reduced the dependence of Env expression on Vpr. A proposed model suggests that the interaction of MR with Env is detrimental to Env stability, although the underlying mechanism remains to be investigated. Previous studies on the interaction of Env with CD4 found that such complexes are actually highly stable and trapped in the ER [132]. Thus, it is not immediately obvious why the interaction of Env with MR should lead to the degradation of Env. Finally, transient expression of MR in 293T cells does not induce Vpr-dependent changes in Env expression (our own unpublished data), suggesting that there may be additional factors involved in the Vpr-dependent stabilization of Env in macrophages. Indeed, as discussed below, Tet2 was implicated in Vpr-mediated enhancement of HIV-1 Env processing and viral infectivity in macrophages [133].

## 6. Vpr-Induced Cell Cycle Arrest and Apoptosis.

Vpr causes G2/M arrest in cycling cells, but the underlying mechanism, as well as the functional importance to the virus, remain controversial [17,67,119,134,135,136,137,138,139,140,141,142,143,144,145]. Indeed, the ability to cause cell cycle arrest is conserved among various primate Vprs [78,146]. Yet, its benefit(s) to HIV replication has remained unclear. The reason is, at least in part, that many cell types, in which Vpr induces G2/M arrest, do not exhibit a significant replication defect in the absence of Vpr. Also, the ability of Vpr to induce G2/M arrest is cell-type specific. For instance, SIV_AGM_ and SIV_SYK_ Vpr are able to induce G2/M arrest in African green monkey cells but not in human cells whereas HIV-1, HIV-2, and SIV_SM_ Vpr proteins function in both simian and human cell types [147]. Nevertheless, it has been reported that cells in G_0_ and early G1 phase are not permissive to HIV replication, whereas cells in the G2 phase are permissive [148]. This is true not only for terminally differentiated macrophages, which rarely divide [149] but also for cells that were arrested in G2 by gamma-irradiation [148]. Such cells are able to continue protein synthesis but cannot proceed into mitosis [148]. One cellular target of Vpr that presumably is involved in cell cycle progression is human Vpr-interacting protein (hVIP), also referred to as Mov34. Mov34 was identified more than twenty years ago as a Vpr interacting factor through a yeast two-hybrid screen [150]. However, follow-up studies examining its biological function are scarce and not recent [151,152]. Mov34 is a member of the eIF3 family, a large multimeric complex that regulates transcriptional events and is essential for G1/S and G2/M phase progression through the cell cycle [152]. Cell cycle progression is regulated by cyclin-dependent kinases (CDK) whose expression level varies throughout the cell cycle and whose activity is controlled by interaction with cyclins and by phosphorylation/dephosphorylation of specific residues in the respective CDK (reviewed in [153]). Progression of cells from the G2 to M phase requires the interaction of the CDK p34^cdc2^ with cyclin B. Cyclin B levels are high late in the G2 phase leading to dephosphorylation of inhibitory residues on p34^cdc2^, thus triggering progression to M phase. The Vpr-induced G2/M arrest was shown to involve a reduction in p34^cdc2^ kinase activity, which was associated with increased phosphorylation at two inhibitory phosphorylation sites on p34^cdc2^ [154]. The same study also showed that mutants of Vpr that were defective for inducing G2/M arrest failed to inhibit p34^cdc2^/cyclin B kinase activity. Conversely, the expression of constitutively active p34^cdc2^ mutants inhibited Vpr-induced G2/M arrest [154]. How exactly Vpr accomplishes this task remains unclear. Interestingly, the induction of both cell cycle G2/M arrest and apoptosis by virion-associated Vpr requires viral entry but not viral replication, since reverse transcriptase and protease inhibitor treatments do not prevent these Vpr effects [71,155].

More recent studies suggest that G2/M arrest encompasses factors involved in DNA damage control [35,36,37,38,63,156,157]. In support of that, silencing of DCAF1 or expression of Vpr mutants unable to bind DCAF1, abolished G2/M arrest [35,36,37,38,63]. The requirement of DCAF1 for Vpr-induced G2/M arrest suggests that it involves the proteasomal degradation of one or more cellular factors. Vpr also induces host-cell apoptosis, and this effect has been linked to cell-cycle arrest [158] but was also reported to be caused by direct effects of Vpr on mitochondrial membrane permeability [159]. Induction of apoptosis does not require de novo Vpr synthesis [160]. The effect of Vpr on apoptosis is inhibited in cells expressing high levels of the host factor gelsolin, which competes with Vpr for binding to the voltage-dependent anion channel (VDAC) [161]. Finally, it has been shown that Vpr induces autophagy in the infected CD4+ T cells. Experiments performed on THP-1 cells indicate that autophagy might play a role in the cellular response to virus infection in macrophages as well [162].

## 7. Vpr-Induced Protein Degradation

Genetic studies designed to measure the in vivo mutation rate of HIV-1 revealed that the mutation rate in the absence of Vpr was up to four times higher than in Vpr expressing virus [19]. This effect was attributed to the interaction of Vpr with DNA repair enzymes uracil DNA glycosylase (UNG2 [163,164]) and the UV excision repair enzyme RAD23A [121]. The human UNG gene is expressed by alternative splicing as a mitochondrial form (UNG1) and a nuclear form (UNG2) [165]. Nuclear UNG2, as well as the related uracil deglycosylase SMUG1, were subsequently shown to be degraded by Vpr via a proteasomal degradation pathway [66]. In the absence of Vpr, both UNG2 and SMUG1 are packaged into viral particles [66,166] and it was argued that excision of uracil from HIV-1 cDNA, which can be created by APOBEC3G-mediated deamination of cytosine residues or by mis- incorporation of dUTP during reverse transcription could lead to fragmentation of the viral genome, thereby exacerbating the antiviral effect of APOBEC3G [66].

Thus, preventing virion incorporation of UNG2 or SMUG1 through Vpr-mediated proteasomal degradation could be beneficial to viral fitness, especially under conditions of low cellular dNTP pools that could promote misincorporation of dUTP. In the course of studying effects of Vpr on the cell cycle, a Vpr-associated E3 ubiquitin ligase complex was identified consisting of Cul4, DDB1 (damaged DNA binding protein 1), DDA1 (DDB1 and DET1 associated 1), and VprBP (also referred to as DCAF1) [35]. The formation of this complex was required for Vpr-induced G2/M arrest. Furthermore, recombinant UNG2 was found to specifically interact with Vpr to form a heterotrimeric complex with DCAF1 in vitro as well as in vivo [34]. Recently, the crystal structure of the DDB1-DCAF1-Vpr-UNG2 complex was solved [41]. The structure of the complex is schematically shown in Figure 3. Structural analysis of this complex revealed that Vpr binds DCAF1 in a manner similar to Vpx despite limited sequence similarity, i.e., primarily through the N-terminal tail and helix α3 [41]. The same study also revealed that Vpr uses structural mimicry to bind UNG2 through its DNA binding domain. Thus, like Vif, Vpu, and Vpx, the HIV-1 Vpr protein has evolved as an adaptor molecule that functions by connecting a cellular E3 ubiquitin ligase complex to cellular substrates resulting in their ubiquitination and subsequent proteasomal degradation.

## 8. Proteins Targeted by Vpr

Early studies on Vpr relied on yeast-two-hybrid assays [167] or phage display screening [168] for the identification of Vpr-interacting cellular proteins such as UNG2 [169]. Technological advances have made it possible to study changes in whole cellular proteomes in response to viral infection or other external factors. Indeed, a recent study compared total proteomes of uninfected CEM-T4 T cells to cells infected with WT or Vpr-defective HIV-1 [53]. The authors observed depletion of previously reported Vpr targets, including HLTF [39,40], ZGPAT [46], MCM10 [137], UNG [66], TET2 [43], and MUS81 and EME1 [42,119]. Furthermore, they observed the depletion of DCAF1. DCAF1 is a component of the E3 ubiquitin ligase complex used by Vpr to degrade cellular targets and had been previously identified as a target of Vpr [170]. In addition to these known targets of Vpr, the study identified almost 2000 other cellular proteins whose expression was affected by HIV-1 in the presence of Vpr [53]. Some of the Vpr interacting factors with promising functions are listed in Table 2. Of note, this global effect of HIV-1 Vpr was not observed for HIV-2 Vpr. Surprisingly, the number of proteins downmodulated following infection with Vpr-defective virus was very small. Also, a DCAF1 binding defective Vpr mutant (Q65R) was almost completely inactive indicating that regulation of cellular proteins by Vpr requires the assembly of the DCAF1/DDB1/Cul4 E3 ubiquitin ligase complex. Overall, Vpr was necessary and sufficient to cause significant depletion of at least 302 proteins and upregulation of 413 [53]. Of the 302 proteins depleted by Vpr, more than 80% resided in the nucleus. Interestingly, of the Vpr targets known to be involved in G2/M arrest (i.e., MCM10, MUS81, and EME1), only depletion of MCM10 correlated with G2/M arrest in CEM-T4 cells in the proteomics study [53]. However, the study identified three additional factors (SMN1, CDCA2, and ZNF267) whose depletion phenocopied Vpr-induced cell cycle arrest. One possible weakness of this study is that it was done in a cell-type that does not require Vpr for efficient virus replication [171]. Thus, only time will tell, which (if any) of the Vpr-responsive factors identified above is the true target that needs to be depleted by Vpr to allow for replication in terminally differentiated macrophages.

## 9. Does Vpr Deplete HDAC to Sustain Active LTR-Driven Virus Production?

As noted in the introduction, Vpr was implicated in the regulation of HIV latency based on the fact that serum Vpr was able to activate virus expression from resting PBMC from HIV-infected individuals [124,125] as well as the observation that CTIP2, a cellular factor associated with chromatin-modifying enzymes, is degraded by Vpr [126]. Intriguingly, Vpr was found to also deplete class I histone deacetylases (HDACs) on chromatin [45,172]. Histone deacetylases are enzymes involved in the regulation of gene expression. Histone acetylation is normally found on actively transcribed genes (reviewed in [173]). Indeed, hyperacetylation of histones on the HIV-1 LTR was found to be correlated with the active transcription of HIV-1 provirus [174], whereas hypoacetylation of those histones was correlated with HIV-1 latency [175]. Importantly, inhibition of HDACs by HDAC inhibitors was found to reactivate latent HIV-1 [176]. This observation has stimulated efforts to employ HDAC inhibitors as a means to purge the viral reservoir [177]. As far as natural reactivation of HIV-1 from latency is concerned, Vpr was found to deplete all members of the class 1 HDAC family without significantly affecting the other HDAC classes [45,172]. However, even in a Vpr overexpression system, depletion of HDAC I proteins was never complete suggesting a presence of a thus far undefined regulatory mechanism. Nevertheless, these results suggest that Vpr plays a role in sustaining an active LTR and maintain active virus production in infected macrophages [45].

## 10. Vpr Activates SLX4 to Promote Cell Cycle Arrest and May Help the Virus Escape from Immune Sensing

SLX4 is a protein involved in DNA repair. Its proposed function is to process DNA intermediates and to act in crosslink repair through interaction with SLX1, XPF-ERCC1, and Mus81-EME1 [178,179]. The SLX4 complex is a SUMO E3 ligase that SUMOylates SLX4 itself and the XPF subunit of the DNA repair/recombination XPF-ERCC1 endonuclease [180]. Binding of Vpr was shown to cause premature activation of the SLX4 complex and to promote cell cycle arrest [119]. Knockdown of any component of the SLX4complex resulted in the failure of Vpr to cause cell cycle arrest [42]. It is important to note that Vpr-induced activation of the SLX4 complex is not a consequence of G2/M arrest but precedes cell cycle arrest [181]. Cell cycle arrest involves Vpr-induced activation of the DNA damage surveillance proteins ataxia-telangiectasia-mutated kinase (ATM) and ATM and Rad3-related kinase (ATR) that detect DNA lesions and trigger downstream signaling cascades [182,183]. Interestingly, the interaction of Vpr with SLX4 does not result in proteasomal degradation of SLX4 but leads to ubiquitination of Mus81 and hyperphosphorylation of EME1, which may result in the digestion of viral DNA and help to escape immune sensing [181,184]. It is unclear how SLX4 might discriminate intact viral DNA that could lead to productive infection from defective DNA that needs to be destroyed. One hypothesis states that only short DNAs generated by reverse transcription errors may be sensitive to nucleases [181]. While the potential contribution of Vpr towards escape from immune sensing is an important function, the effects of Vpr on SLX4 are unlikely to explain the reported restriction of Vpr-defective HIV-1 in differentiated macrophages and the rather subtle restriction reported for T cell lines.

## 11. Vpr-Induced Degradation of the DNA Repair Endonuclease Mus81 Is Genetically Separable from Vpr-Induced Cell Cycle Arrest. What Is the Benefit for HIV-1?

Mus81 is a DNA repair endonuclease that plays an essential role in the completion of homologous recombination during DNA repair events. Mus81 facilitates rapid DNA replication and mediates cellular responses to exogenous replication stress [185]. Mus81 forms a complex with its cofactor EME1, which binds SLX4 and SLX1 to form a four-subunit complex [42]. It was reported that Vpr directly binds to SLX4 and induces an untimely activation of the SLX4-associated Mus81-Eme1 complex through PLK1-mediated phosphorylation of EME1 [119]. Vpr caused a DCAF1-dependent decrease in Mus81 levels while levels of SLX4 seemed to be unaffected by binding to Vpr [119]. However, degradation of Mus81 is probably independent of the Vpr function to cause G2/M arrest. In fact, Mus81 degradation is genetically separable from Vpr’s ability to induce G2/M arrest [52].

## 12. Vpr-Induced Degradation of HLTF Is Genetically Separable from Vpr-Induced Cell Cycle Arrest

Helicase-like transcription factor (HLTF) HLTF is the human ortholog of yeast Rad5 and has roles in transcription, chromatin remodeling, DNA damage repair, and tumor suppression (reviewed in [186]). In conjunction with SHPRH, it coordinates different forms of postreplication DNA repair [187]. It also plays a role in the G2/M transition and apoptosis in brain cells [188]. Two recent studies independently identified HLTF as a target of HIV-1 Vpr [39,40]. In one study, a proteomics approach was used to identify novel Vpr-interacting proteins [40]. The second study employed stable isotope labeling by amino acids (SILAC). This analysis identified HLTF as the only protein among 2000 analyzed, whose expression in Vpr-expressing cells was significantly reduced [39]. Both studies confirmed that Vpr caused degradation of HLTF through a DCAF1-dependent proteasomal pathway. Because of its known role in regulating G2/M transition, it was conceivable that Vpr affects HLTF to exploit its function in the cell cycle arrest via the interaction with SLX4. However, both studies found that Vpr-mediated degradation of HLTF is independent of G2/M arrest but appears to precede this arrest and is genetically separable from G2/M arrest. Interestingly, HIV-2 Vpr was unable to induce degradation of HLTF [40]. Both in HIV-1 infected T cells, and in primary macrophages, Vpr was found to down-regulate HLTF [39]. However, siRNA-mediated silencing of HLTF in macrophages did not alleviate the Vpr restriction of these cells [39], suggesting that HLTF is not the (only) factor whose expression in macrophages restricts the replication of Vpr-defective HIV-1. Further evidence that HLTF is not solely responsible for the restriction of Vpr-defective virus in macrophages comes from the observation that replication of HIV-2 in human macrophages is Vpr-dependent, yet, HIV-2 Vpr does not deplete HLTF. More recently, it was reported that Vpr counteracts HLTF-mediated restriction of HIV-1 infection in T cells [189]. The investigators used a pairwise replication competition assay (PRCA) rather than the usual parallel replication studies to measure the relative replication fitness of WT versus Vpr-defective virus in Jurkat cells. Under such conditions, HLTF was found to restrict the replication of Vpr-defective virus. However, silencing of HLTF and ExoI, either alone or in combination, only partially reversed the Vpr-defect suggesting that inhibition of replication of Vpr-defective HIV-1 in terminally differentiated macrophages likely involves additional factors and may in fact be a multifactorial process.

## 13. Vpr Induces Degradation of Exo1 but the Benefits to Viral Replication Are Unclear

Exo1 is a member of the Rad2 family of nucleases that in vitro was shown to possess 5′->3′ exonuclease and 5′-flap endonuclease activities (for review see [190]). Exo1 plays a role in homologous recombination during meiosis. It also plays a role in nucleotide excision repair. Exo1 was identified as a target of Vpr in the course of a focused screen of postreplication DNA repair proteins [191]. Vpr was shown to induce degradation of Exo1 in a proteasome-dependent but cell cycle-independent manner. Interestingly, si-RNA-mediated depletion of Exo1 did not phenocopy the Vpr-induced cell-cycle arrest, suggesting that Exo1 depletion by Vpr does not fully explain the Vpr-induced G2/M cell cycle arrest [191]. As discussed in the section on HLTF above, silencing of HLTF and ExoI, either alone or in combination, only partially reversed the Vpr-defect suggesting that inhibition of replication of Vpr-defective HIV-1 in terminally differentiated macrophages may involve additional factors [189].

## 14. Vpr Induces Degradation of the Transcriptional Repressor ZIP: What Is in It for HIV-1?

ZIP, also known as ZGPAT, is a transcriptional repressor that was recently identified as a factor repressing the EGFR oncogene expression and suppressing breast carcinogenesis [192]. ZIP functions by recruiting the nucleosome remodeling and deacetylase complex. A splice variant, sZIP, was shown to antagonize the repressor function of ZIP [193]. Both ZIP and sZIP were recently identified as Vpr interacting proteins [46]. In fact, both proteins were found to be degraded by Vpr in a DCAF1-dependent manner in transiently transfected HeLa cells. Expression of ZIP or sZIP does not appear to be limited to macrophages, and it does not appear to have a role in Vpr-mediated G2/M arrest. Thus, the functional relevance of ZIP or sZIP degradation by Vpr is currently unclear.

## 15. Vpr-Induced Degradation of APOBEC3G: Vif is Not Impressed

APOBEC3G is a cytidine deaminase with potent antiviral activity [194]. APOBEC3G was identified as the major target of Vif, which induces its proteasomal degradation thus enabling HIV-1 to replicate in APOBEC3G-expressing cells [195]. In the absence of Vif, APOBEC3G is packaged into virus particles and can cause deamination-induced mutations in the viral genome during reverse transcription. Deamination of cytidine residues produces deoxyuridine, which can be read as threonine residues during second-strand synthesis, resulting in G to A mutations in the plus strand. In addition, the presence of deoxyuridine residues in viral cDNA may trigger cellular DNA repair mechanisms that could lead to fragmentation of the viral genome. The presence of Vif during particle production effectively blocks the encapsidation of APOBEC3G both by proteasomal degradation as well as through degradation-independent mechanisms [194]. More recently, Vpr was implicated in the proteasomal degradation of APOBEC3G [47]. Depletion of APOBEC3G by Vpr required DCAF1 but was only observed in transient transfection studies. Indeed, the infection of A3.01 cells, which express sub-lethal levels of APOBEC3G, allows for replication of Vif-defective virus expressing Vpr [196]. However, only Vif-expressing WT virus but not Vif-defective virus reduced levels of endogenous APOBEC3G during the course of infection (our own unpublished data). These results suggest that the effect of Vpr on APOBEC3G expression under physiological conditions is very limited.

## 16. Vpr Degrades Tet-2 to Promote Virus Production by Increasing Expression of IL-6

DNA can be modified by methylation of cytosine (resulting in 5-methylcytosine (5mC). 5-mC methylation is a type of epigenetic modification known to affect cellular transcription, X-chromosome inactivation, and genomic imprinting and was until recently thought to be irreversible [197]. The identification of Tet (Ten eleven translocation) proteins led to the realization that Tet2 is a member of the family of DNA dioxygenases, which catalyze iterative oxidation of 5-mC, eventually leading to demethylation of cytosine [198]. Inactivation of TET enzymes was shown to cause developmental defects and hematopoietic malignancies [199]. Interestingly, regulation of Tet2 activity involves an interaction with DCAF1 that in normal cells, induces mono-ubiquitination of Tet2 and promotes Tet2 binding to chromatin [199]. In contrast, expression of Vpr in HIV-infected cells induces poly-ubiquitination of Tet 2, resulting in its proteasomal degradation [43]. This leads to increased HIV-1 replication through a mechanism that is unrelated to Vpr-induced G2/M cell cycle arrest [43]. Instead, Tet2 was found to actively repress IL-6 expression by recruiting HDAC to the IL-6 promoter [200]. Vpr-mediated degradation of Tet was shown to increase the production of IL-6 in HIV-infected macrophages which was associated with increased virus production, as shown by the sensitivity of this effect to IL-6-neutralizing antibodies [43]. In addition to its effect on IL-6 expression, Tet2 was shown to affect interferon-induced transmembrane protein 3 (IFITM3) expression by reducing the demethylation of the IFITM3 promoter [133]. Indeed, it was shown that macrophages constitutively express IFITM3 in a Tet2-dependent manner. IFITM3 together with IFITM2 was previously reported to restrict HIV-1 infection by antagonizing the HIV-1 Env glycoprotein processing and virion incorporation [201]. Expression of Vpr was shown to enhance Env processing and packaging by degrading Tet2, which in turn reduces IFITM3 expression. Thus, reduced IFTIM3 expression enhanced Env processing and virion infectivity and sustained IL-6 expression to increase HIV-1 replication.

## 17. Conclusions and Outlook

Vpr is a viral accessory protein that plays a critical role in virus replication in vivo. Characterization of the physiologically relevant function(s) of Vpr has been hampered by the lack of convenient cell systems in which to test Vpr phenotypes. Nevertheless, many novel Vpr interacting factors were described over the course of recent years. However, for most of these factors, their precise mode of action relevant to Vpr function remains incompletely understood. It will be important to assess the physiological relevance of all the Vpr targets in relevant cell types. Despite recent progress in gene silencing and gene knockout techniques, working with primary human cells, and in particular, MDM, remains tedious. Thus, a focus of future research will have to be to improve technical procedures for working with MDM. In addition, systematic screening for additional Vpr interacting factors, especially proteins expressed primarily in macrophages, will be required to further our understanding of the role of Vpr in the virus life cycle.

## Figures and Tables

**Figure 1 cells-08-01310-f001:**
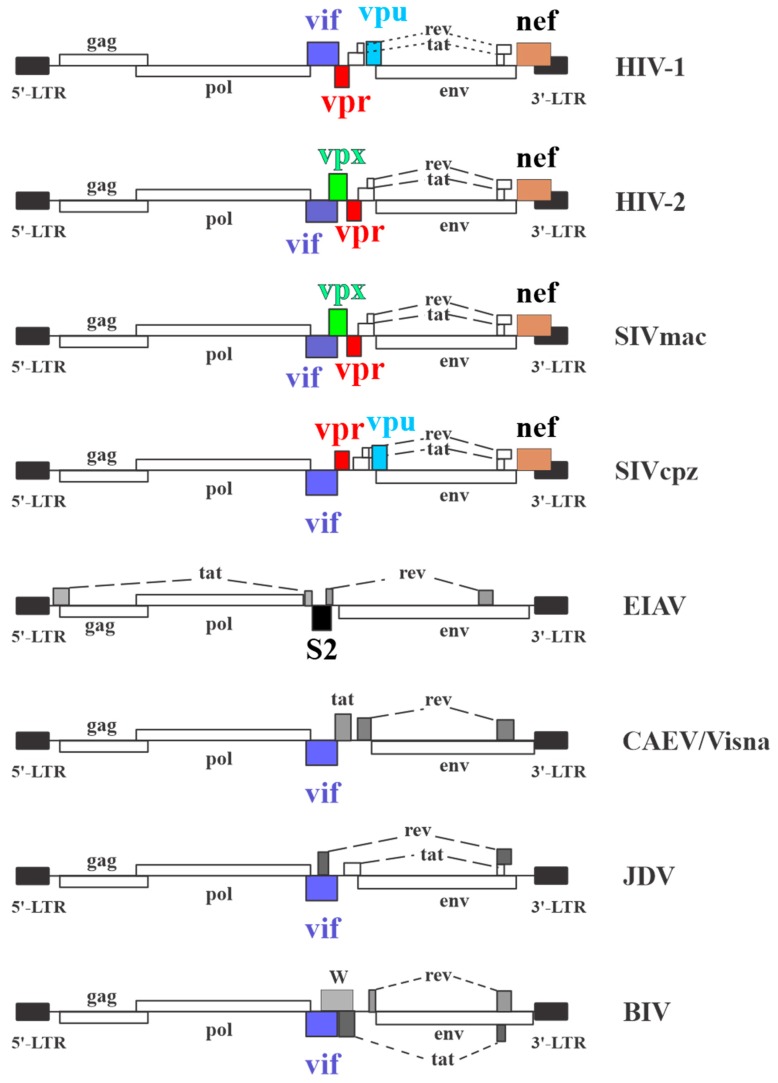
Lentiviral genome organization. Lentiviral genomes share significant similarities with respect to the location of the structural *gag*, *pol*, and *env* genes. However, they differ in number and type of accessory proteins. All but EIAV encode a *vif* gene, and all primate lentiviruses (i.e., human and simian immunodeficiency viruses) encode *vpr* and *nef* genes, whereas *vpu* and *vpx* genes are limited to HIV-1 and HIV-2 and related viruses, respectively. HIV = Human Immunodeficiency Virus, SIV = Simian Immunodeficiency Virus, EIAV = Equine Infectious Anemia Virus, CAEV = Caprine Arthritis Encephalitis Virus, JDV = Jembrana Disease Virus, BIV = Bovine Immunodeficiency Virus.

**Figure 2 cells-08-01310-f002:**
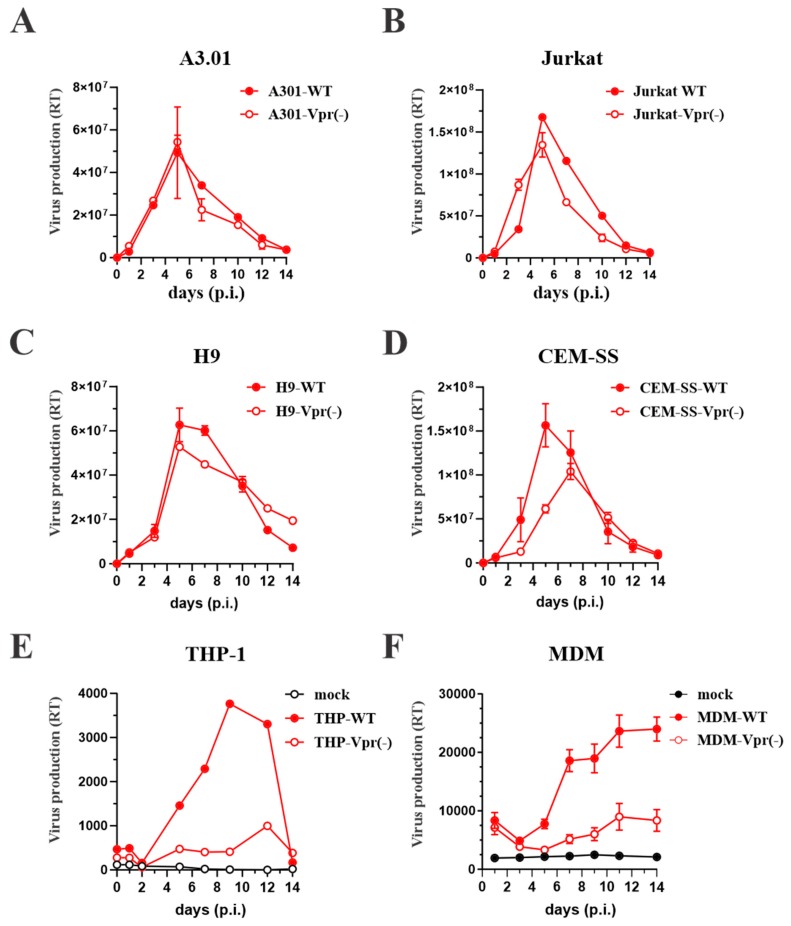
Replication of wild type (WT) and Vpr-defective (Vpr(−) HIV-1 in various cell types. A3.01, Jurkat, H9, and CEM-SS are human CD4+ T cell lines that support the replication of X4-tropic HIV-1 (e.g., NL4-3). THP-1 is a human monocytic cell line and MDM are primary human monocyte-derived macrophages. Terminally differentiated MDM and differentiated THP-1 cells are susceptible to infection by R5-tropic HIV-1 (e.g., AD8). NL43wt, NL43vpr(−), AD8wt, and AD8vpr(−) virus stocks were prepared by transfecting corresponding molecular clones into HEK293T cells. Virus-containing supernatants were harvested 24 h after transfection, filtered, and concentrated by centrifugation through a 20% sucrose cushion. Pelleted viruses were suspended in RPMI medium, adjusted to equal reverse transcriptase activity and used to infect the above cell lines. Culture supernatants were collected for 14 days in 2–3-day intervals and virus-associated reverse transcriptase activity was determined and plotted as a function of time. We found that wild type and Vpr-defective NL43 viruses replicated with very similar kinetics in all four T cell lines tested (panels **A**–**D**) suggesting that Vpr is not critical for virus replication in these cells. In contrast, replication of Vpr-defective AD8 virus in THP-1 cells and in MDM was severely restricted (panels **E**,**F**) demonstrating the importance of Vpr for replication in these cells.

**Figure 3 cells-08-01310-f003:**
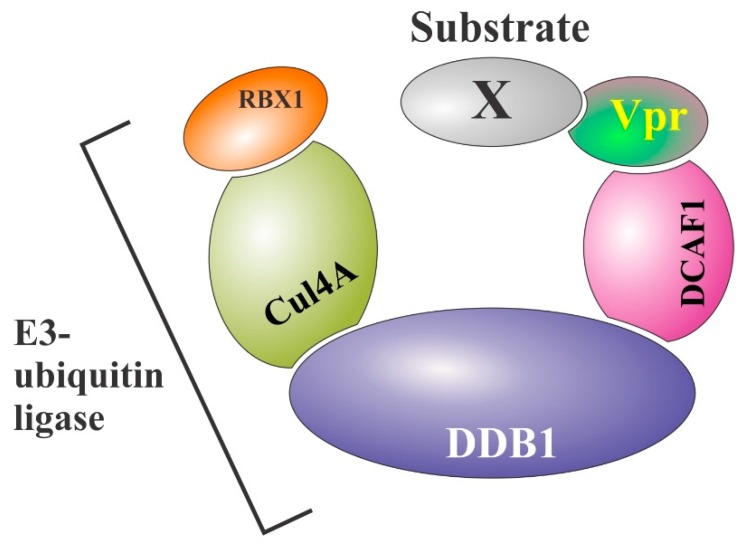
Schematic structure of the CRL4DCAF1 E3 ubiquitin ligase complex assembled by Vpr for the ubiquitination and subsequent degradation of Vpr-interacting substrates (X). Binding of cellular factor X (e.g., HLTF, Exo1, Mus81, ZIP, Tet2, and other thus unidentified substrates) to Vpr induces its poly-ubiquitination through the Vpr-associated E3 ubiquitin ligase complex, which marks these proteins for subsequent degradation by the cellular proteasome machinery.

**Table 1 cells-08-01310-t001:** Vpr mutants and their functional phenotypes.

Mutant	Effect	Citation
**T19A**	Associated with long-term non-progressor (LTNP) phenotype	[32]
**L23F**	Nucleoporin binding mutant, diffuse nucleocytoplasmic distribution, unable to cause G2/M arrest	[56]
**E24R, R36P**	Selectively disrupts Vpr binding to helicase-like transcription factor (HLTF) without disturbing the UNG2 degradation	[40] [57]
**K27M**	Nucleoporin binding mutant, diffuse nucleocytoplasmic distribution, no G2 arrest, no apoptosis induction, no accumulation at the nuclear envelope	[58]
**A30F**	Unable to bind gag and package into particles	[8]
**F34I**	Decreased Vpr incorporation into virions compared to WT Vpr Unable to bind the nuclear envelope but causes G2/M arrest	[59] [60]
**P35N**	Unable to interact with cyclophilin A	[61]
**R36W**	Associated with RP phenotype	[32]
**Y50A**	G2/M arrest defective mutant	[58]
**W54R**	Defective for UNG2 loading onto CRL4 but retains binding to DCAF	[34,62]
**G56A**	Defective for HLTF degradation, ability to arrest cell cycle maintained	[39]
**Q65R**	DCAF (also known as VprBP) binding mutant. Reduced cell-to-cell spread of HIV-1 from macrophages to CD 4+ T lymphocytes	[63] [64]
**L68M**	Associated with rapid progressor (RP) phenotype	[32]
**L68A**	Nuclear export mutant	[65]
**F69A**	DCAF binding mutant, but associates with Exo1 like wild type Vpr	[41]
**H71R**	DCAF binding mutant, unable to induce G2/M cell cycle arrest Decreased Vpr incorporation into virions	[66] [59,67]
**F72L**	Decreased nuclear import of the virus; LTNP	[68]
**R73S**	G2/M arrest defective mutant with decreased apoptosis induction ability Transcriptional activation	[69] [70]
**R77Q**	Associated with LTNP phenotype, impaired induction of apoptosis	[32,71,72]
**S79A**	G2/M arrest defective mutant, unable to activate TAK1	[39,73]
**R80A**	Decreased ability to cause cell cycle arrest	[74]
**R85Y**	Associated with rapid disease progressor phenotype	[32]
**R90K**	Does not induce cell cycle arrest, no apoptosis, localizes to the nuclear envelope	[56]
**R90N**	Associated with LNTP phenotype	[32]
**R85QRR**	Impaired nuclear targeting by the virus	[75]

**Table 2 cells-08-01310-t002:** Table of Vpr interacting partners.

Vpr interacting partner	Function	Citation
**DCAF (VprBP)**	Part of the E3 ubiquitin ligase complex used by Vpr to degrade cellular targets	[120]
**Importin**	Involved in nuclear translocation of the preintegration complex	[202]
**SNF2h**	Binding to chromatin	[203]
**Adenine nucleotide translocator**	a mitochondrial ATP/ADP antiporter, interaction with Vpr induces apoptosis	[204]
**Cyclophilin A**	Enhancing virus replication	[205]
**VDAC**	Permeabilization of mitochondrial membrane	[206]
**Nucleoporin hCG1**	Docking of Vpr at the nuclear envelope	[85]
**DNA repair protein HHR23A**	Controversial function in G2/M-arrest	Structure reported in [207]
**DNA damage binding protein (DDB1)**	E3 ubiquitin ligase complex assembly	[208]
**HLTF**	Role in DNA repair, degraded by Vpr	[57]
**Exo1**	Role in DNA repair, degraded by Vpr	[191]
**SLX4 complex**	G2/M cell cycle arrest	[139]
**SMUG1**	UNG2 and SMUG1 initiate base excision repair mechanisms.	[50]
**ZIP and sZIPI**	Degraded by Vpr, interaction with DCAF, function unknown	[46]
**Mus81**	Possible function in G2/M arrest	[52]
**TET2**	Sustaining IL-6 expression in macrophages	[43,133]
**Uracil DNA glycosylase (UNG2)**	G2/M arrest independent repair of DNA containing uracil residues	[163] [62]
	Decrease of the mutation rate in HIV-1, packaging the uracil DNA glycosylase into viral particles, and hence, enabling replication in primary cells	[209] [166]

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
