# Peer review of "Vpr and Its Cellular Interaction Partners: R We There Yet?"

_cells, 2019, doi:10.3390/cells8111310_

Round 1

Reviewer 1 Report

In this review authors summarize the state of art research on lentiviral accessory protein vpr. This is a timely and much needed review from one of the experts on the topic. In addition to summarizing the past literature, which the authors have done meticulously, it would have been good to see a paragraph about where the field is headed and which studies may help to resolve the outstanding questions.

Specific points

Fig 2: Authors should indicate which cell lines are T and which are monocytic etc. Same reference has been cited twice #119 and #126 “Vpr overcomes macrophage-specific restriction of HIV-1 Env expression and virion production”

Reviewer 2 Report

This is a very thorough review about HIV vpr. It is well written and very informative. However, the manuscript as it stands could benefit from a few improvements. First, in regards to the resolution of figures. Also, abbreviations should all be defined at first appearance in the text. In Figure 2, the cellular origin of all cell lines that are stated should be defined. Finally, sub-titles for sections 9-16 are insufficient and should be more descriptive.

Reviewer 3 Report

In this manuscript, Drs. Fabryova and Strebel provided a thorough review on the interaction of Vpr with host factors. This is a good reference covering most important topics regarding the protein. The text is written in beautiful English and with a good logical structure, and therefore comfortable to read. I believe this manuscript would be highly worth publication. Please improve the clarity of the text where possible. My comments are listed below.

The authors are certainly more interested in HIV-1 Vpr than Vpr proteins of HIV-2 and SIV. However, the term Vpr seems to be used in different meanings in the text. So, it could be helpful to better clarify what Vpr stands for at some specific points. Alternatively, the authors could declare early in the text that “Vpr” stands for “HIV-1 Vpr” unless otherwise specified. For example: In page 2 line 49, “virus particles would contain about 700 Vpr molecules”. This applies to HIV-1 but not to HIV-2 (40-50 copies). For reference, Kewalramani VN, Park CS, Gallombardo PA, Emerman M. Protein stability influences human immunodeficiency virus type 2 Vpr virion incorporation and cell cycle effect. Virology. 1996 Apr 15;218(2):326-34. In page 4 line 97, “Vpr (Viral Protein R)” is a 96 amino acid protein …”. I guess this is HIV-1 Vpr. In Table 1, “Vpr mutants …”. This seems to be HIV-1 Vpr. It could be meaningful to indicate the similarity of Vpx (HIV-2/SIVsm/SIVmm) to Vpr (SIVagm), though the information might not be essential for the review. For reference, Sharp PM, Bailes E, Stevenson M, Emerman M, Hahn BH. Gene acquisition in HIV and SIV. Nature. 1996 Oct 17;383(6601):586-7. In Introduction (page 1), the gene names should be in italic. In page 3 lines 65-67, “... the function of Vpr has been the fact that its contribution to viral replication in commonly used experimental tissue culture systems such as immortalized T cell lines is generally modest”. Do the authors mean that the Vpr effect in T-cell lines is limited relative to the Vpr effect in monocyte cell lines? Please describe more clearly. Figs are in miserably low DPIs. Figure legends should be provided as texts (not as part of images) and should be much longer so that readers can understand the figures without referring to the text. Fig 1 should be much larger and clearer. I am not sure whether HIV-1, HIV-2. SIVmac, and SIVcpz should come in this order rather than in an order considering the homogeneous groups (HIV-1/SIVcpz and HIV-2/SIVmac). It might be meaningful to show the SIVagm genome to indicate that some SIVs do not have vpu or vpx. Fig 2 also seems to be too small even in a higher resolution. It should be much larger. As the figure represent experiment results, the legend should describe the experiment so that the figure should be comprehensible. All the six cell lines used in the experiment should be clearly explained. It should be clarified which are T cell lines and which are not, as well as what was the key finding. Moreover, it is essential to clarify the source of the data. If the data was replicated from a previous publication, then the authors need permission for the reuse of the data. In Fig. 3, I would recommend to update the information so that the figure and the legend can be comprehensible to even those who are not familiar with Ubiquitin signaling. The two tables should be numbered.
